# Protective Effect of [Cu(NN_1_)_2_](ClO_4_) Complex in Rainbow Trout Challenged against *Flavobacterium psychrophilum*

**DOI:** 10.3390/microorganisms10112296

**Published:** 2022-11-19

**Authors:** Maialen Aldabaldetrecu, Mick Parra, Sarita Soto-Aguilera, Pablo Arce, Amaya Paz de la Vega Quiroz, Rodrigo Segura, Mario Tello, Juan Guerrero, Brenda Modak

**Affiliations:** 1Laboratory of Coordination Compounds and Supramolecularity, Faculty of Chemistry and Biology, University of Santiago of Chile, Av. Bernardo O’Higgins 3363, Santiago 9170022, Chile; 2Laboratory of Natural Products Chemistry, Centre of Aquatic Biotechnology, Faculty of Chemistry and Biology, University of Santiago of Chile, Av. Bernardo O’Higgins 3363, Santiago 9170022, Chile; 3Laboratory of Bacterial Metagenomic, Centre of Aquatic Biotechnology, Faculty of Chemistry and Biology, University of Santiago of Chile, Santiago 9170002, Chile; 4Laboratory of Electroanalysis, Department of Materials Chemistry, Faculty of Chemistry and Biology, University of Santiago of Chile, Av. Bernardo O’Higgins 3363, Santiago 9170022, Chile

**Keywords:** *Flavobacterium psychrophilum*, copper (I) complex, antibacterial activity, coumarin, rainbow trout

## Abstract

Previously, we reported an in vitro evaluation regarding antibacterial effects against *F. psychrophilum* by a new Cu (I) complex, [Cu(NN_1_)_2_](ClO_4_). This study presents the results of an in vivo evaluation of [Cu(NN_1_)_2_](ClO_4_) added as a dietary supplement against *F. psychrophilum* in rainbow trout. The results showed that the administration of [Cu(NN_1_)_2_](ClO_4_) at 29 and 58 µg/g of fish for 15 days does not affect the growth of rainbow trout. On the other hand, the amount of copper present in the liver, intestine, and muscle of rainbow trout was determined. The results showed that the amount of copper in the liver, when compared between treated fish and control fish, does not change. While, in the intestine, an increase in the fish fed at 58 µg/g of fish was observed. In muscle, a slight decrease at 29 µg/g was obtained. Additionally, copper concentrations in the pond water after 15 days of feeding with the [Cu(NN_1_)_2_](ClO_4_) complex showed the highest levels of copper. Finally, the effect of the administration of [Cu(NN_1_)_2_](ClO_4_) for 15 days at 58 µg/g of fish was evaluated against *F. psychrophilum*, where a 75% survival was obtained during 20 days of challenge.

## 1. Introduction

Aquaculture is one of the fastest growing sectors of the food industry worldwide, supplying more than 50% of the total fish and shellfish needed for human consumption, and with 157 million tons produced in 2020 [1]. However, this rapid progress has brought with it overproduction conditions, such as high population densities in fisheries, which increases the risk of outbreaks of infectious diseases produced by several pathogens [2,3].

Among the pathogens that seriously affect aquaculture production is *Flavobacterium psychrophilum*, a Gram-negative bacteria, which is the causative agent of “cold water bacterial disease” (BCWD) and “rainbow trout fingerling syndrome” (RTFS) in a large number of freshwater fish species worldwide [4,5,6]. In fish, this bacteria causes external infections such as skin erosions and ulcers, dark pigmentation and rot in the fins and gills [5,7], erratic swimming behavior, and spiral movements observed in juvenile fish [7]. Infection by this bacterium constitutes an important problem in the aquaculture industry, causing great economic losses in the global aquaculture of freshwater salmonids [4,8,9]. *F. psychrophilum* affects the majority of salmonid species; however, rainbow trout (*Oncorhynchus mykiss)* are one of the most susceptible to the disease [10]. This species is particularly important in Chile, since from 2009 to 2018, Chile has led as the world’s leading producer of rainbow trout, with a total of 1,542,935 tons.

The control method most used for *F. psychrophilum* infection is the use of antibiotics, of which the most used are erythromycin, florfenicol, oxytetracycline, and oxolinic acid [8,9]. However, the excessive use of antibiotics in aquaculture worldwide has caused problems due to the development and spread of bacterial resistance, generating a danger to food safety, as well as environmental problems [11]. Several reports have shown that antibiotics promote the selection and propagation of a large and diverse group of genes associated with antimicrobial resistance. This facilitates the horizontal transfer of these genes between aquatic and terrestrial bacteria, causing health risks when transferred to human pathogens. [12,13,14]. In the case of *F. psychrophilum*, an increased resistance to oxytetracycline has been observed and a percentage of resistance to this antibiotic ranging from 46% to 80% has been reported [15,16,17,18,19]. This has caused an increase in the persistence of *F. psychrophilum*, giving rise to frequent bacterial outbreaks, with a significant increase in mortality in fish farms, causing considerable economic losses [5].

Therefore, there is has been an increased interest in identifying new therapeutic alternatives to antibiotics. In this regard, medicinal plants have been shown as a natural source of many active components with effective antimicrobial properties against bacterial pathogens [20,21]. On the other hand, food supplements based on functional compounds with antibacterial capabilities have aroused the interest of researchers due to the effects these may have on the treatment of bacterial infections affecting fish [22]. Various studies have shown the effect of food additives with protective effects against various pathogens. For example, the administration of Quercetin in zebrafish (*Danio rerio*) improved immunity by increasing resistance against Aeromonas salmonicida [23]. Additionally, the administration of caffeic acid in Nile tilapia (*Oreochromis niloticus*) also improved the immune response by increasing resistance against Aeromonas veronii [24].

Similarly, the antibacterial effect of copper coordination complexes with natural or synthetic ligands has been tested on pathogenic bacteria, such as *Escherichia. coli, Pseudomonas aeruginosa*, and *Staphylococcus aureus*, and has been observed to have better effect than their separate components [25,26].

On the other hand, it has been observed that the incorporation of copper in the diet of aquatic organisms contributes towards improving the functioning of various physiological factors, which allows a better response to infections caused by bacterial pathogens [27]. For example, it has been shown that the administration of a copper-nicotinate complex protects croakers against *Yersinia ruckeri* by stimulating the cellular and humoral immune response [28].

Although copper is an important trace element present in vertebrates, it is required in all organisms for a number of key physiological and biochemical functions related to growth, development, and metabolism [27]. It is essential to maintain its concentration at adequate levels so as not to generate adverse effects on the body, either due to a lack or an excess of copper [29]. On the other hand, copper complexes with natural ligands with antibacterial capacity have been proposed as new alternatives to conventional antibiotic treatments for bacterial diseases [30,31].

Regarding the above, we previously reported the synthesis of a new Cu(I) complex, [Cu(NN_1_)_2_](ClO_4_), where NN_1_ is an imine ligand 6-((quinolin-2-ylmethylene)amino)-2H-chromen-2-one, a derivative of natural compound coumarin, which has been shown to have a better antibacterial effect in vitro than its precursors, coumarin and copper (I) salt [24]. However, the in vivo effect that this copper (I) complex could have against an infection of *F. psychrophilum* has not been determined.

Accordingly, in the present study, we present the results obtained from an evaluation of the antibacterial effects of the copper (I) complex added as a dietary supplement against *F. psychrophilum* infection in rainbow trout. In addition, copper concentrations in different organs of the fish fed with the complex and in the water of the ponds where these fish were developed were determined.

## 2. Materials and Methods

### 2.1. Complex (I) Copper

Copper (I) complex [Cu(NN_1_)_2_]ClO_4_ (Figure 1) was obtained in four synthetic steps. The first step is the nitration of coumarin, a molecule of natural origin, used as the base of the final compound. The second step is the reduction to obtain the amino coumarin. In the third step, the ligand described as NN_1_ is obtained by a condensation reaction with 2-quinolinecarboxaldehyde. Finally, in the last step, the complex is obtained by reaction of 2 NN_1_ ligands with the precursor [Cu(CH_3_CN)_4_]ClO_4_. The specific reaction conditions, as well as the characterization of the complex by NMR techniques, UV–Vis and cyclic voltammetry, are described in our previous work [32].

### 2.2. Fish and Maintenance

Pre-smolt rainbow trout (*O. mykiss*) weighing between 15–30 g were used (Fish farming Federico Albert Taupp Rio Blanco, Los Andes, Chile). The fish were acclimated for one week before treatment at 12 °C in freshwater aquariums with a biomass of 14 g/L, continuous aeration, and fed with commercial pellets (EWOS MICROTM 2 mm, Cargill, Coronel, Chile) at 1% of body weight. The fish were maintained in freshwater with a pH between 6.6 and 7, the salinity was adjusted to 6 PSU with NaCl to prevent fungal infection, and total ammonia was maintained in a range below 0.02 mg/L. Around 70% of the water in all the aquariums was changed every day before feeding. Water parameters were monitored daily prior to and after changing the water. Feeding, changing the water, and measuring water parameters were all manually conducted. The fish were maintained in accordance with the ethical standards of the Institutional Ethics Committee of the Universidad de Santiago de Chile and the current relevant legislation. The authorization of the Ethics Committee of the Universidad de Santiago de Chile to perform experiments with fish in the project FONDECYT 1180265 was granted on 26 January 2018, and is consigned in the report Nº42.

### 2.3. Effect of the [Cu(NN_1_)_2_](ClO_4_) Complex on the Growth of Rainbow Trout

The effect of the administration of the [Cu(NN_1_)_2_](ClO_4_) complex on the growth of rainbow trout was evaluated, for which a total of 80 rainbow trout of 30 ± 2 g were used. The fish were divided into 4 different groups with 10 fish per pond in duplicate (total *n* = 20). Group A: control (untreated). Group B: vehicle control. Group C: fish treated with 29 µg/g of fish with the complex (I) copper. Group D: fish treated with 58 µg/g of fish with the [Cu(NN_1_)_2_](ClO_4_) complex. Group A fish were fed with commercial pellets, while Group B fish were fed with commercial pellets mixed with edible oil. The fish in groups C and D were treated with [Cu(NN_1_)_2_](ClO_4_) complex emulsified with edible oil to improve their adherence to food pellets. The treatments were administered for 15 days and the fish were weighed and measured before the start of acclimatization and at the end of the experiment.

The compound amount administered per day (mT) was obtained using Equation (1), considering m as the average fish mass (g fish), and C corresponds to the concentrations administered as previously described. The average mass of the fish was corrected every week.
(1)mT=C × m (g fish)

Fish growth was evaluated by calculating the specific growth rate (SGR) and weight gain (Equations (2) and (3)).
(2)SGR=(ln final weight−ln initial weight)×100/days
(3)Weight gain=final weight−initial weight

### 2.4. Determination of the Copper Concentration in Fish Tissues and Water

A total of 160 rainbow trout of 15 ± 2 g were used, divided into 4 different groups with 20 fish per pond in duplicate (total *n* = 40). Group A: control (untreated). Group B: vehicle control. Group C: fish treated with [Cu(NN_1_)_2_](ClO_4_) complex 29 µg/g of fish. Group D: fish treated with complex (I) copper 58 µg/g of fish. Group A fish were fed commercial pellets, while Group B fish were fed commercial pellets mixed with edible oil. The fish in groups C and D were treated with [Cu(NN_1_)_2_](ClO_4_) complex emulsified with edible oil to improve their adherence to food pellets.

Water samples were collected from all ponds before the start of feeding (T0), 1 h post-feeding (T1), 24 h post-feeding and before the water change (T2), 1 h post-feeding on day 5 of treatment (T3), day 6 of treatment and before the water change (T4), 1-h post-feeding on day 10 of treatment (T5), day 11 of treatment and before the water change (T6), 1 h post-feeding on day 14 of treatment (T7), and day 15 of treatment and before the water change (T8).

The water samples were passed through a sterile 33 mm/0.45 μm PVDF filter and the copper concentration was determined using Graphite Furnace Atomic Absorption (AAS-GF) equipment (Thermo Scientific, Model iCE3000 Series, Waltham, MA, USA). This equipment was fully automated so the calibration was directed by software using a 50 µL copper master standard, with which the calibration curves were constructed. The calibration curves were between values of 2–10 µL and 5–30 µL, with an adjustment or correlation coefficient of one min of r^2^ = 0.990. All measurements were made in triplicate.

After 15 days of treatment, the fish were killed by benzocaine overdose (30 mg/L); liver, intestine, and muscle tissue were removed from each fish and stored at −80 °C until use. For the analysis of the organs, pools of 5 fish were made, obtaining a *n* = 10 per treatment for each organ. The treatment of samples of the different rainbow trout tissues was carried out by measuring up to 0.60 g of tissue and by means of closed acid digestion using a microwave equipment model Ethos Easy, Milestone Srl brand; a complete digestion of the sample was achieved, according to the Protocol 3052 for the acid digestion assisted by microwaves for organic and siliceous matrices of the Environmental Protection EPA (Washington, DC, USA). This protocol involves the digestion of the sample using 9 mL of nitric acid (HNO_3_) and 1 mL of hydrofluoric acid (HF), reaching a temperature of 180 °C in 20 min and then maintaining this temperature for a further 10 min. The samples were then volumetric to 25 mL using Milli-Q water and then measured by means of a Thermo Scientific Atomic Absorption Furnace (AAS-GF), Thermo Scientific brand (Model iCE3000 Series). The calibration curves used were between the values of 5–30 µL and 15–40 µL, with an adjustment or correlation coefficient of one min of r^2^ = 0.990. All measurements were made in triplicate.

### 2.5. Bacterial Strain and Growth Conditions

The *F. psychrophilum* isolate NBRC 100250 (ETECMA), originally isolated from coho salmon (*Oncorhynchus kisutch*), was used for the challenge experiment. The isolate was cultivated in TYES broth (pH = 7.2) at 15 °C for 72 h with stirring at 180 rpm.

### 2.6. Rainbow Trout Challenged with F. psychrophilum

The ability of the [Cu(NN_1_)_2_](ClO_4_) complex to protect rainbow trout against *F. psychrophilum* NBRC 100250 was evaluated. For this experiment, 60 rainbow trout of 15 ± 2 g were used, divided into 3 different groups with 10 fish per pond in duplicate (total *n* = 20). Group A: control (untreated). Group B: challenge control. Group C: fish treated with [Cu(NN_1_)_2_](ClO_4_) complex 58 µg/g of fish. The group of treated fish were fed for 7 days with [Cu(NN_1_)_2_](ClO_4_) complex mixed with commercial marigold oil homogenized by mechanical action with commercial pellets; later, they were challenged with 1 × 10^8^ CFU of *F. psychrophilum* NBRC 100250 by intraperitoneal injection (previously anesthetized with 30 mg/L of benzocaine). Subsequently, the fish were fed for a further 8 days with 58 µg/g of [Cu(NN_1_)_2_](ClO_4_) complex mixed with commercial marigold oil homogenized by mechanical action with commercial pellets. The second group of fish were only challenged with 1 × 10^8^ CFU of *F. psychrophilum* NBRC 100250 by intraperitoneal injection (previously anesthetized with 30 mg/L of benzocaine), as the challenge control. Daily mortality was recorded for 20 days post-challenge.

Survival analysis was evaluated using Kaplan-Meier estimate. The comparison between survival curves was performed using the Mantel-Cox non-parametric test (log-rank) with a *p* < 0.05. The surviving fish were sacrificed with 30 mg/L of benzocaine. Kidneys were collected from dead and surviving fish and stored at −80 °C for analysis.

### 2.7. DNA Extraction and qPCR

In addition, the bacterial load in the dead and surviving fish was quantified, for which genomic DNA was extracted from fish kidneys using a Wizard^®^ Genomic DNA Purification Kit (Promega, Madison, WI, USA), following the manufacturer’s instructions. Total DNA was then quantified by UV-Vis spectrophotometry using a Tecan INFINITE M200 Pro and stored at −20 °C until use. Posteriorly, an absolute quantification was performed in 96-well plates (AXIGEN) using a Stratagene Mx 3000P (Agilent Technology, Santa Clara, CA, USA), for which the DNA was adjusted to a concentration of 50 ng/µL. The PCR mix contained 0.5 µL of forward and reverse primers (10 µM) of gen rpoC of *F. psychrophilum* [25], 5 µL of Sensimix SYBR NO-ROX (Bioline USA, Tauton, MA, USA), 3 µL of ultra-pure H_2_O (Corning), and 1 µL of genomic DNA. The thermal profile used was 95 °C for 3 min, 35 cycles of 95 °C for 10 s, 60 °C for 20 s, and 72 °C for 20 s. The Ct value was used in the calibration curve to obtain the number of copies present in the sample.

### 2.8. Statistical Analysis

Statistical analysis was performed using GraphPad Prism 5 program, using a nonparametric Mann-Whitney *t*-test analysis with a *p* < 0.05.

## 3. Results

### 3.1. Effect of the [Cu(NN_1_)_2_](ClO_4_) Complex on the Growth of Rainbow Trout

The effect of 15 days of administration of two concentrations of [Cu(NN_1_)_2_](ClO_4_) complex, 29 and 58 µg/g of fish, as a food supplement on the growth of rainbow trout was evaluated. During feeding, no negative effect was observed on trout behavior, such as erratic swimming, lack of appetite, or mortality, in the fish treated with both concentrations of [Cu(NN_1_)_2_](ClO_4_) complex, as well as in the vehicle control (data not shown). To evaluate the effect on the growth of the fish, the specific growth rate (SGR) and the weight gain were determined. In addition, the fish were measured before and after administration of the complex. The SGR calculation did not show statistically significant differences between the fish without treatment (C = 1.545 ± 0.347%) and the treated fish with 29 µg/g of fish (29 = 1.574 ± 0.429%) and 58 µg/g of fish (58 = 1.647 ± 0.241%), or with the vehicle control (V = 1.328 ± 0.635) (Figure 2a). Similar results were observed in the calculation of weight gain, where no differences were observed between the fish without treatment (C = 7.842 ± 2.079 g) and the treated fish with 29 µg/g of fish (29 = 8.129 ± 2.769 g) and 58 µg/g of fish (58 = 8.417 ± 1.687 g), or with the vehicle control (V = 6.996 ± 3.733 g) (Figure 2b). On the other hand, no difference in the length of the fish was observed at the end of the experiment between the treatments (Figure 2c). These results show that the administration of the [Cu(NN_1_)_2_](ClO_4_) complex in the two concentrations tested for 15 days does not affect rainbow trout growth.

### 3.2. Determination of the Copper Concentration in Rainbow Trout Organs after 15 Days of Feeding with [Cu (NN_1_)_2_](ClO4) Complex

The amount of copper present in the liver, intestine, and muscle of the rainbow trout was determined after 15 days of feeding with [Cu(NN_1_)_2_](ClO_4_) complex (Figure 3). The results showed that there was no difference in the amount of copper present in the liver between the control fish and the fish treated with commercial oil (vehicle control) or the fish treated with both concentrations of the [Cu(NN_1_)_2_](ClO_4_) complex (29 and 58 µg/g of fish) (Figure 3). On the other hand, when the intestine of the fish was evaluated, it was possible to detect a statistically significant increase in the amount of copper between the fish fed with 58 µg/g of fish with the [Cu(NN_1_)_2_](ClO_4_) complex (10.0 ± 3.4 ppm) and the control fish (6.6 ± 1.1 ppm), but not with fish treated with 29 µg/g of fish with the [Cu(NN_1_)_2_](ClO_4_) complex (7.2 ± 1.9 ppm). Finally, when the muscle tissue of the fish was evaluated, a statistically significant decrease in the amount of copper was only observed between the fish treated with commercial oil (0.4 ± 0.07 ppm) and the fish fed with 29 µg/g of fish with the [Cu(NN_1_)_2_](ClO_4_) complex (0.5 ± 0.07 ppm) compared to the control fish (0.6 ± 0.2 ppm).

### 3.3. Determination of the Copper Concentration in Pond Water after 15 Days of Feeding with [Cu(NN_1_)_2_](ClO_4_) Complex

The experiments were carried out in ponds without a recirculation system, in which 70% of the water was changed each day. To measure the concentration of copper in the water, samples were collected at different times after the start of the feeding. The samplings were carried out 1 h post-feeding on day 1, 5, 10, and 14 of experimentation, and 24 h post-feeding corresponding to days 2, 6, 11, and 15 of experimentation, before carrying out the water exchange on each of the corresponding days.

The results showed that 1 h after the start of the feeding (T1), an increase was observed in the copper concentration present in the water of the fish treated with both concentrations of the [Cu(NN_1_)_2_](ClO_4_) complex (4.68 ± 0.07 ppm and 5.26 ± 0.47 ppm, respectively) compared to the control fish (2.94 ± 0.77 ppm) and the fish treated with the vehicle (2.56 ± 0.03 ppm). 24 h post-feeding and before the water exchange, corresponding to day 2 of experimentation (T2), similar results were observed, with an increase in the amount of copper present in the water of the fish treated with both concentrations of the [Cu(NN_1_)_2_](ClO_4_) complex (2.36 ± 0.02 ppm and 3.49 ± 0.09 ppm, respectively) compared to the control fish (0.78 ± 0.13 ppm) and the fish treated with the vehicle (1.21 ± 0.02 ppm). On day 5 of treatment, 1 h post-feeding (T3), the water of the fish treated with both concentrations of the [Cu(NN_1_)_2_](ClO_4_) complex presented once again higher copper concentrations than both controls. However, 24 h post-feeding on day 6, before the water exchange (T4), only the water present in the fish treated with the highest concentration of the [Cu(NN_1_)_2_](ClO_4_) complex showed higher copper levels (21.79 ± 12.12 ppm) than the control fish (6.60 ± 0.96 ppm) and the treated fish with the vehicle (4.17 ± 0.72 ppm).

When copper concentrations in the water were measured 1 h after feeding on day 10 of treatment with [Cu(NN_1_)_2_](ClO_4_) complex (T5), an increase in the copper concentration was observed in the water of the fish treated with both concentrations of the [Cu(NN_1_)_2_](ClO_4_) complex compared to both controls. Similar results were observed 24 h after feeding and before water exchange, corresponding to day 11 of treatment (T6). Finally, in the last two analyzed time periods, corresponding to 1 h post-feeding on day 14 of treatment (T7) and 24 h post-feeding and prior to water exchange on day 15 of treatment (T8), the results were similar to what had been observed throughout the experiment. The highest levels of copper were found in the water of the fish treated with concentrations of the compound compared to the control. However, in T8, the highest levels of copper were found; 28.19 ± 7.51 ppm in the water of the fish treated with 29 µg/g of fish with the [Cu(NN_1_)_2_](ClO_4_) complex and 34.05 ± 1.15 ppm in the water of the fish treated with 58 µg/g of fish with the [Cu(NN_1_)_2_](ClO_4_) complex (Table 1).

### 3.4. Effect of the Administration of [Cu(NN1)_2_](ClO_4_) Complex in Rainbow Trout against Challenge with F. psychrophilum

The effect of the administration of the [Cu(NN_1_)_2_](ClO_4_) complex for 15 days, at a concentration of 58 µg/g of fish, was evaluated against infection with *F. psychrophilum*. The results obtained showed that it was possible to obtain a higher survival percentage in the fish treated with [Cu(NN_1_)_2_](ClO_4_) complex versus the fish without treatment (75 and 35%, respectively) during the 20 days of the experiment. Survival analysis was performed using Kaplan-Meier estimation, employing the log-rank test (Mantel-Cox) and the Gehan-Breslow-Wilcoxon test, which obtained a statistically significant difference with a value of *p* = 0.0118 (Figure 4a). Subsequently, the bacterial loads of *F. psychrophilum* present in the dead and surviving fish were analyzed to determine if there was any difference between the control fish and the fish fed with [Cu(NN_1_)_2_](ClO_4_) complex. The results obtained showed that in the dead fish there was no difference in the bacterial loads of *F. psychrophilum* between the fish fed with [Cu(NN_1_)_2_](ClO_4_) complex and the control fish. On the other hand, in the surviving fish, it was not possible to detect differences in the bacterial load of *F. psychrophilum* present between the fish fed with [Cu(NN_1_)_2_](ClO_4_) complex and the control fish. However, it was possible to observe a greater number of fish in which the bacterial load of *F. psychrophilum* was not detected between the fish fed with [Cu(NN_1_)_2_](ClO_4_) complex (n:12) and the control fish (n:4) (Figure 4b). These results show that feeding with the [Cu(NN_1_)_2_](ClO_4_) complex for 15 days is capable of reducing the mortalities caused by *F. psychrophilum* isolate NBRC 100250 in rainbow trout, and is effective in eliminating the bacteria from the fish.

## 4. Discussion

Food supplements based on functional compounds with antibacterial capabilities have aroused the interest of researchers due to the effects they may have on the treatment of bacterial infections affecting fish in aquaculture [26]. In this regard, it has been observed that copper incorporated into the diet of various aquatic organisms contributes to the better functioning of various physiological factors that allow the individual to respond better to bacterial pathogen infections [27]. Although copper is an important trace element present in the body of vertebrates, required for a number of key physiological and biochemical functions in all organisms related to growth, development, and metabolism [27], it is essential to maintain its concentration at adequate levels in order to avoid generating adverse effects in the individual’s body, whether due to a lack of or excess copper [28]. On the other hand, Cu(I) complexes with natural ligands with antibacterial capacity have been proposed as new alternatives to conventional antibiotic treatments for bacterial pathogenic diseases [29,30]. Therefore, in this study, we evaluated the effect of feeding the [Cu(NN_1_)_2_](ClO_4_) complex to rainbow trout, in order to determine effects on fish size and physiological parameters, as well as survival from flavobacteriosis infection.

The first results showed that the administration of the [Cu(NN_1_)_2_](ClO_4_) complex at concentrations of 29 and 58 µg/g of fish for 15 days did not affect the growth of rainbow trout, with no changes observed in the SGR index, weight gain, and length of the animal between the treated fish and the control fish. Our results were similar to other experiments with rainbow trout fed with 15 µg/g of fish (CuSO_4_) for 35 days, where they did not present significant differences in the growth of the fish compared to the controls [27]. On the other hand, in tilapia, it has been observed that the administration of copper into the diet, in concentrations of 16 to 32 mg/kg, does not negatively affect growth, but the administration of concentrations between 4000 and 6000 mg/kg does negatively affect growth [31]. Other positive effects in the administration of diets with CuSO_4_, CuMet (copper-methionine), or CuONano (nano-copper oxide) have been reported on the growth of other species produced, such as the Russian sturgeon [32]. The difference in the results of all these studies suggests that the effect of Cu in the diet depends on different parameters, such as species, life stage, concentration, and type of copper, and probably the composition of the diet.

Another important parameter to analyze was the retention of copper in different tissues of the fish, because an increase in physiological levels is related to different disorders in aquatic animals [27]. After 15 days of feeding with the [Cu(NN_1_)_2_](ClO_4_) complex, no increase in copper concentration was observed in the liver of the fish in any of the treatments. Similar results were reported in red sea bream (*Pagrus major*) fed for 60 days with copper nanoparticles + Vitamin C [33], and in Nile tilapia fed for 60 days with Cu_2_SO_4_ at 16, 24, and 32 mg/kg. However, with fish fed with 4000 and 6000 mg/kg of Cu_2_SO_4_, an increase in copper concentration was found, physiologically affecting the fish. These results show that copper levels sufficient for the requirement of the fish are positive; however, a considerable increase is toxic and worsens the growth and health of the fish [31]. Regarding the amount of copper present in fish muscle, our results did not show an increase in the accumulation of copper in the doses used, similar to that reported in red sea bream (*Pagrus major*) fed with copper nanoparticles [33,34]. Although muscle is not an active tissue in the accumulation of metals, the analysis of the accumulation of copper becomes relevant for human health, since it is the consumable part of the fish. In a study carried out with Atlantic salmon, it was determined that the minimum copper requirement for consumable meat in a diet not supplemented with copper should be 3.5 mg/kg, and in the case of a supplemented diet, it should be between 5–10 mg/kg [35]. Therefore, the doses tested in this study would allow safe levels of copper for human consumption. Finally, when the accumulation of copper in the intestine was analyzed, a greater accumulation was observed in the fish fed with the highest concentration tested in this work (58 µg/g of fish) compared to the control fish. These results are similar to those previously reported in rainbow trout [36]. It has been determined that the intestine is the most active site of absorption and/or concentration of Cu [36]. On the other hand, future studies are necessary to evaluate the effect of increased copper concentrations in the intestine of rainbow trout.

Another relevant indicator is the copper concentrations detected in the aquarium water. In our study, after feeding the fish with copper complex in both concentrations, an increase in copper levels was observed in the aquarium water after the first hour of feeding on days 1, 5, 10, and 14. This may be related to the increase in excretion by the fish after being fed due to the effect of digestion and, therefore, this increase could be due to the copper present in the feces of the fish, increasing the amount of copper in the water of the aquarium. The accumulation of copper in aquatic fish farming systems must be constantly monitored and analyzed to avoid problems regarding the health of animals; when exceeding certain levels of copper the water becomes toxic for fish, affecting various physiological functions caused by oxidative damage generated by the production of high levels of the damaging hydroxyl radical associated with the presence of Cu+ [37]. Due to the fact that we did not observe negative effects on the growth of the fish fed with the concentrations of the [Cu(NN_1_)_2_](ClO_4_) complex evaluated, we can assume that this increase is not considerable enough to affect the fish; however, it is necessary to carry out more experiments at the physiological level to verify this hypothesis. On the other hand, the maximum free copper permitted in the water of fish farms should not exceed 1.7 g/L [26], while the maximum detected in our study was approximately 34 mg/L in the aquariums of the fish fed with 58 µg/g of the [Cu(NN_1_)_2_](ClO_4_) complex. However, more studies are needed to assess whether these values are comparable.

Finally, when evaluating whether this new copper (I) complex is capable of generating protection in rainbow trout against a infection with *F. psychrophilum*, it was observed that fish fed with 58 µg/g fish increased the survival of rainbow trout in comparison with the control. In addition, it was observed that the treatment was also able to eliminate the pathogen in some of the surviving fish. Similar results were obtained in a study in catfish that were fed for two weeks with additional CuSO_4_ at 0, 40, and 80 mg/kg of diet and infected with *F. columnare*; an increase in fish survival in a similar dose-dependent way was observed [38].

The results obtained in this study showed that this new copper (I) complex is capable of generating protection in rainbow trout against *F. psychrophilum* when it is administered for 15 days in the diet at a concentration of 58 µg/g of fish, without generating toxicity in fish or affecting their growth, and without accumulation in liver or muscle tissue.

## 5. Conclusions

We previously synthesized the copper (I) complex [Cu(NN_1_)_2_]ClO_4_ and reported a significant increase in the in vitro antibacterial effect of this new copper (I) complex against an isolate of *F. psychrophilum*, compared to its precursors.

In this study, the in vivo protective effect of the copper (I) complex [Cu(NN_1_)_2_]ClO_4_ against *F. psychrophilum* infection in rainbow trout was evaluated. Results showed that the administration of the copper complex (I) at a concentration of 58 µg/g of fish for 15 days effectively inhibited the mortality caused by *F. pyschrophilum*.

Along with decreasing mortality, the fish fed with the copper (I) complex were not affected in terms of their growth. In addition, there was no significant increase in copper concentration in the muscle or liver of the fish compared to the control fish.

These results allow us to consider this copper (I) complex to be a promising alternative for the treatment of diseases caused by *F. psychrophilum*, allowing us to reduce the high amounts of antibiotics currently used.

Studies related to elucidating the antibacterial mechanism of the [Cu(NN_1_)_2_]ClO_4_ complex should continue to be carried out, as well as its protective effect against other marine pathogens.

Finally, the potential therapeutic use of the complex in fish is very important because there is a great need for safer and effective drugs that can be used in the treatment of infectious diseases caused by current and emerging bacteria, especially when efficient therapies are not available.

## 6. Patents

The results of this work are patented by the INAPI under the title: “Compound that comprises a complex of coordination of Cu(I) with coumarine ligands, [Cu(I)(NN_1_)_2_](ClO_4_), NN_1_: benzopyran-2-one, and [Cu(MeCN)_4_]ClO_4_; Composition and/or food that understand and their use to treat the disease produced by *Flavobacterium psychrophilum* in fish”. Number 2020-000932.

## Figures and Tables

**Figure 1 microorganisms-10-02296-f001:**
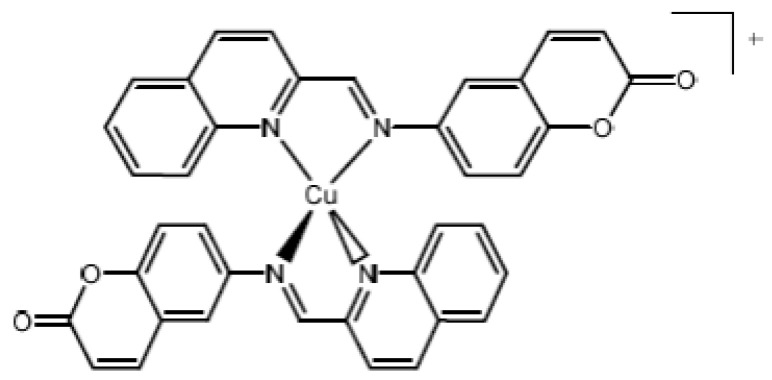
Chemistry structure of copper (I) complex [Cu(NN_1_)_2_]ClO_4_. NN_1_ is 6-((quinolin-2-ylmethylene) amine)-2H-chromen-2-one.

**Figure 2 microorganisms-10-02296-f002:**
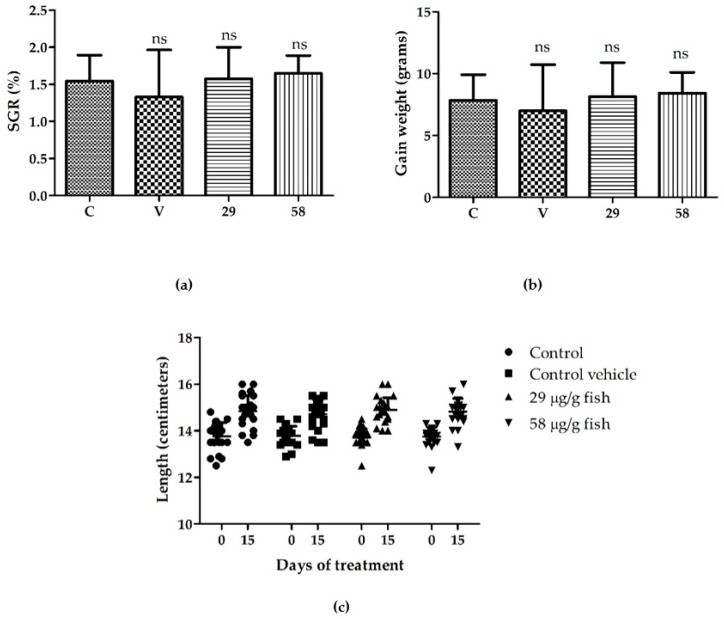
Growth performance of rainbow trout fed with and without [Cu(NN_1_)_2_](ClO_4_) complex supplemented diet. C = control, V = vehicle control, 29 = 29 µg/g of fish, and 58 = 58 µg/g of fish: (**a**) % SGR; (**b**) gain weight, (**c**) fish length at the beginning and end of the experiment. Statistical analysis nonparametric Mann-Whitney *t-*test with a *p* < 0.05, ns indicates no significant difference.

**Figure 3 microorganisms-10-02296-f003:**
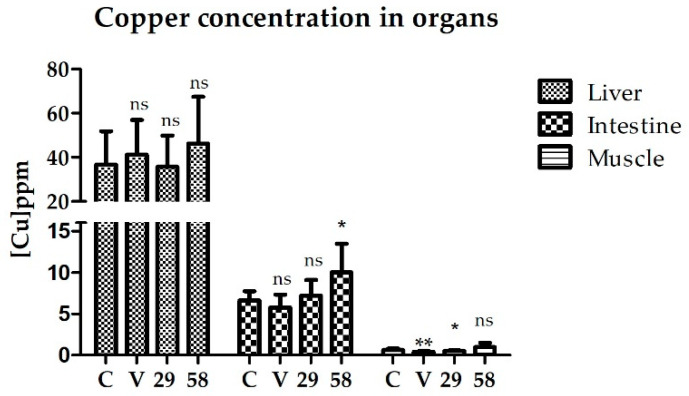
Copper concentration determined in the liver, intestine, and muscle of rainbow trout after 15 days of feeding with [Cu(NN_1_)_2_](ClO_4_). C = control, V = vehicle control, 29 = concentration 29 µg/g of fish, and 58 = concentration 58 µg/g of fish. Statistically significant differences were determined in respect to the control by a one-way non-parametric *t*-test (Mann–Whitney) (* *p* < 0.05, ** *p* < 0.01)y, ns = not significant.

**Figure 4 microorganisms-10-02296-f004:**
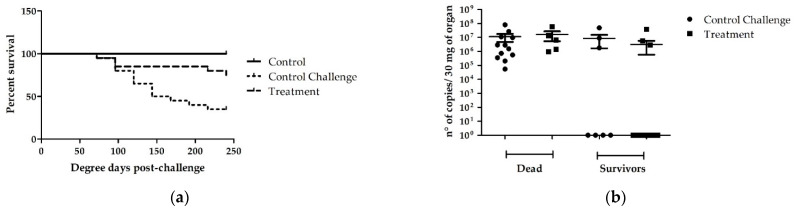
Effect of the administration of [Cu(NN_1_)_2_](ClO_4_) complex in the rainbow trout diet against infection with *F. pyschrophilum*: (**a**) Percentage of survival, (**b**) bacterial loads. Statistically significant differences were determined by comparing the curves of treatment versus challenge control, using a long-rank test (Mantel-cox) and Gehan-Breslow-Wilcoxon test (*p* < 0.05).

**Table 1 microorganisms-10-02296-t001:** Copper concentration in pond water at different times during feeding with [Cu(NN1)_2_](ClO_4_) complex. Different letters mean statistically significant differences between treatments. Statistical analyses were independently carried out at each time analyzed. Statistically significant differences were determined compared to the control by a one-way nonparametric *t*-test (Mann–Whitney) (*p* < 0.05).

Copper Concentration (ppm)				
Time	Control	C. Vehicle	29 µg/g of Fish of Complex	58 µg/g of Fish of Complex
T0	1.94 ± 0.63 a	1.52 ± 0.89 a	1.17 ± 0.46 a	1.41 ± 0.02 a
T1	2.94 ± 0.77 a	2.56 ± 0.03 a	4.68 ± 0.07 b	5.26 ± 0.47 b
T2	0.78 ± 0.13 a	1.21 ± 0.02 b	2.36 ± 0.02 c	3.49 ± 0.09 d
T3	2.56 ± 0.28 a	3.54 ± 0.54 b	5.07 ± 0.44 c	6.57 ± 2.5 c
T4	6.60 ± 0.96 a	4.17 ± 0.72 b	7.43 ± 2.42 ac	21.79 ± 12.12 d
T5	1.14 ± 0.23 a	0.42 ± 0.07 b	2.35 ± 0.89 c	6.23 ± 0.88 d
T6	1.41 ± 0.10 a	1.43 ± 0.02 a	12.49 ± 1.80 b	19.60 ± 5.82 c
T7	2.56 ± 0.23 a	2.44 ± 0.22 a	9.56 ± 2.37 b	11.51 ± 0.34 b
T8	3.57 ± 0.43 a	3.80 ± 0.55 a	28.19 ± 7.51 b	34.05 ± 1.15 b

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
