# Peer review of "Protective Effect of [Cu(NN1)2](ClO4) Complex in Rainbow Trout Challenged against Flavobacterium psychrophilum"

_microorganisms, 2022, doi:10.3390/microorganisms10112296_

Round 1

Reviewer 1 Report

Paper “Effect of the administration [Cu(NN1)2](ClO4) complex in rainbow trout in a challenge against Flavobacterium psychrophilum” by Aldabaldetrecu et al. describe the activity of [Cu(NN1)2](ClO4) complex against Flavobacterium psychrophilum. The manuscript was well written and contains promising data, arranged in a scientific manner. However, the manuscript needs English editing before considering publication in a microorganism journal.

Author Response

We appreciate all the positive comments to our work.

According to the reviewer's suggestion, English has been revised

Reviewer 2 Report

This article "Effect of the administration [Cu(NN1)2](ClO4) complex in rain-2 bow trout in a challenge against Flavobacterium psychrophilum" by Aldabaldetrecu et al. While this work could be published, following concerns need to be addressed.

1.      The title should be changed to be contain the effect of the tested compound on rain-2 bow trout.

2.      Positive control should be included in bioactivity assays, otherwise it will be impossible to clearly validate the results.

3.      Table 1: authors should add the p value to legend.

4.       Figure 4: authors should perform statistical analysis.

5.      The results part should be improved.

6.      Discussion section should be more elaborated and discuss the obtained results on rain-2 bow trout with other results.

7.      Conclusion section should be separated and improved and mention the future uses of Cu(NN1)2](ClO4) complex.

8.      References in text should be updated.

Author Response

We appreciate all the positive comments to our work.
A file with the responses to the evaluator's suggestions is attached.

Reviewer 3 Report

Dear Authors! Thank You for an interesting article. Below are a few of my questions and comments:

Lines 53-60

I think the issue of antibiotics could be discussed in more details. Is fish used for food the object of this article? How dangerous is the use of fish’s antibiotics for humans? Perhaps some studies should be given on the harm of antibiotics, if such harm takes place. I believe that the issue of antibiotics and medicines used for food products is very serious and needs additional sources.

Lines 61-62

There is really a great number of solutions to the problem of antimicrobial and antibacterial effect of various additives, feeds, substances. I suggest that the authors strengthen this section with a large number of references, mentioning the mechanisms of action.

Lines 72-78

More data on the substance under study is needed, supported by references from other authors, since link 24 belongs to the authors of this article.

Line 86-87

Although the authors provide a link to their work, where they synthesize the substance under discussion, it is necessary to give more details so as not to burden readers with finding the source.

Lines 123-127

I recommend adjusting the formulas according to the design requirementsFigure 2 is to small. Please make it larger, especially Figure 2 c.

Lines 345-356

This part is more suitable for the Introduction section.

Line 371-373

It is clear that the contribution of a large number of factors has been identified, but it may be possible to name and specify them?

How can the accumulation of copper in the environment of fish affect metabolic processes in the ecosystem? And on production processes?

Author Response

(The authors gave the same response as above.)

Round 2

Reviewer 2 Report

This article has improved considerably in the revision and all of my comments have been addressed.